# Simple Path Structural Encoding for Graph Transformers

**Louis Airale** [1]   **Antonio Longa** [1]   **Mattia Rigon** [1]   **Andrea Passerini** [1]   **Roberto Passerone** [1]

## Abstract

Graph transformers extend global self-attention to graph-structured data, achieving notable success in graph learning. Recently, Relative Random Walk Probabilities (RRWP) has been found to further enhance their predictive power by encoding both structural and positional information into the edge representation. However, RRWP cannot always distinguish between edges that belong to different local graph patterns, which reduces its ability to capture the full structural complexity of graphs. This work introduces Simple Path Structural Encoding (SPSE), a novel method that utilizes simple path counts for edge encoding. We show theoretically and experimentally that SPSE overcomes the limitations of RRWP, providing a richer representation of graph structures, particularly for capturing local cyclic patterns. To make SPSE computationally tractable, we propose an efficient approximate algorithm for simple path counting. SPSE demonstrates significant performance improvements over RRWP on various benchmarks, including molecular and long-range graph datasets, achieving statistically significant gains in discriminative tasks. These results pose SPSE as a powerful edge encoding alternative for enhancing the expressivity of graph transformers.

## 1. Introduction

Graphs are pervasive across diverse domains, representing complex relationships in areas such as social networks (Otte & Rousseau, 2002), molecular structures (Quinn et al., 2017), and citation graphs (Radicchi et al., 2011). Recent advances in graph neural networks (GNNs) have driven significant progress in learning from graph-structured data (Kipf & Welling, 2017; Veličković et al., 2018; Bodnar et al.,

2022; Lachi et al., 2024; Ferrini et al., 2024; Duta & Liò, 2024), yet these models often face challenges in capturing long-range dependencies and structural patterns due to their reliance on localized message passing (Alon & Yahav, 2021; Topping et al., 2022).

Inspired by their success in vision and sequence learning tasks (Vaswani, 2017; Dosovitskiy et al., 2021), transformers have been extended to graph learning problems (Yun et al., 2019; Dwivedi & Bresson, 2020; Ying et al., 2021; Kreuzer et al., 2021). Unlike traditional GNNs, graph transformers leverage global self-attention, allowing each node to attend to all others within a graph, regardless of distance. This flexibility overcomes the limitations of message-passing approaches but introduces new challenges, particularly in designing suitable positional and structural encodings that capture the inherent irregularities of graphs.

For directed acyclic graphs (DAGs), positional encodings (PEs) based on partial orderings can be directly applied (Dong et al., 2022; Luo et al., 2024b; Hwang et al., 2024). However, for general undirected graphs, successful PEs often rely on eigendecompositions of the graph Laplacian, drawing inspiration from sinusoidal encodings in sequence transformers (Dwivedi & Bresson, 2020; Mialon et al., 2021). These approaches encode node-level information but fail to capture the full structural complexity of edge patterns in node neighborhoods.

To address this limitation, several graph transformer architectures incorporate initial message-passing steps to encode local substructures (Wu et al., 2021; Mialon et al., 2021; Chen et al., 2022). While effective, these methods focus solely on node representations and do not exploit the potential of injecting pairwise structural encodings directly into the self-attention mechanism. Recent studies have explored structural edge encodings in pure transformer architectures, based for instance on Laplacian eigenvectors, heat kernels, or shortest path distances (Kreuzer et al., 2021; Ying et al., 2021; Chen et al., 2023). Despite their utility, these encodings are limited in expressivity, particularly for capturing local cyclic patterns or higher-order substructures.

A promising alternative is the use of random walks as a structural encoding method (hereafter RRWP for Relative Random Walk Probabilities following Ma et al. (2023)), whereby richer structural information is encoded at the

---

[1]University of Trento, Trento, Italy. Correspondence to: Louis Airale <louis.airale@unitn.it>, Roberto Passerone <roberto.passerone@unitn.it>.

*Proceedings of the 42ⁿᵈ International Conference on Machine Learning*, Vancouver, Canada. PMLR 267, 2025. Copyright 2025 by the author(s).

edge level by considering random walk probabilities, and which has led to substantial improvements in the performance of state-of-the-art graph transformers (Ma et al., 2023; Menegaux et al., 2023). However, RRWP struggles to differentiate between distinct graph structures in certain cases. Meanwhile, parallel research on message-passing GNNs has demonstrated the benefits of simple paths (or self-avoiding walks) in enhancing model expressivity beyond the 1-Weisfeiler-Leman (WL) isomorphism test (Michel et al., 2023; Graziani et al., 2024). These findings motivate the exploration of simple paths as a structural encoding mechanism in graph transformers.

In this work, we introduce **Simple Path Structural Encoding (SPSE)**, a novel method for structural edge encoding that replaces RRWP in graph transformers. SPSE encodes graph structure by counting simple paths of varying lengths between node pairs, capturing richer structural information than random walks. To address the computational challenges encountered when exact simple path counting is unfeasible, we propose an efficient algorithm based on successive DAG decompositions using depth-first search (DFS) and breadth-first search (BFS). This approach avoids the exponential memory costs of path enumeration, enabling scalability to long path lengths. [1]

We validate SPSE on extensive benchmarks, including molecular datasets from Benchmarking GNNs (Dwivedi et al., 2023), Long-Range Graph Benchmarks (Dwivedi et al., 2022), and Large-Scale Graph Regression Benchmarks (Hu et al., 2021). SPSE consistently outperforms RRWP in graph-level and node-level tasks, demonstrating significant improvements in molecular and long-range datasets. We also characterize limit cases of the algorithm, and identify situations in which the performance might be more sensitive to approximate path counts.

The remainder of this paper is structured as follows. Section 2 introduces key concepts and notations. Section 3 analyzes the limitations of RRWP and motivates SPSE. The proposed path-counting algorithm and encoding method are detailed in Section 4. Finally, experimental results and related works are discussed in Sections 5 and 6, respectively.

## 2. Preliminaries

### 2.1. Graph Theory

Let $\mathcal{G} = (\mathcal{V}, \mathcal{E}, \mathbf{X})$ be a graph, where $\mathcal{V}$ is the set of nodes, $\mathcal{E} \subseteq \mathcal{V} \times \mathcal{V}$ is the set of edges, and $\mathbf{X} \in \mathbb{R}^{|\mathcal{V}| \times d}$ represents the node features of dimension $d$. The *adjacency matrix* $\mathbf{A} \in \{0, 1\}^{|\mathcal{V}| \times |\mathcal{V}|}$ is a square matrix that represents the connectivity of the graph, i.e. $\mathbf{A}_{ij}$ is one if there is an edge

between nodes $i$ and $j$, and zero otherwise. The diagonal degree matrix $\mathbf{D} \in \mathbb{R}^{|\mathcal{V}| \times |\mathcal{V}|}$ is a square matrix where the diagonal element $\mathbf{D}_{i,i}$ represents the degree of node $i$. Formally, $\mathbf{D}_{i,i} = \sum_j \mathbf{A}_{i,j}$ and $\mathbf{D}_{i,j} = 0$ for $i \neq j$.

**Definition 2.1** (Walk). *Given a graph $\mathcal{G}$, a **walk** is a finite sequence of nodes $v_0, v_1, \ldots, v_m$, where each consecutive pair of nodes $(v_i, v_{i+1})$ is connected by an edge, i.e., $(v_i, v_{i+1}) \in \mathcal{E}$. The number of edges in a walk is referred to as the **walk length**.*

**Definition 2.2** (Simple Path). *A **simple path** (here indifferently called **path**), is a walk in which all nodes are distinct. The number of edges in a simple path is called the **simple path length** (or **path length**). Paths themselves constitute graphs called **path graphs**. The **distance** between two nodes is the length of the shortest path between these two nodes.*

**Definition 2.3** (Cycle). *A **cycle** is a walk where all nodes are distinct, except for the first and last ones which are identical. A graph composed of a single cycle is a **cycle graph**.*

**Definition 2.4** (Random Walk Matrix). *Given a graph $\mathcal{G}$ and $k \in \mathbb{N}^*$, the k-hop **random walk matrix** $P_k \in \mathbb{R}^{|\mathcal{V}| \times |\mathcal{V}|}$ gives the landing probabilities of random walks of length $k$ between all pairs of nodes. Its closed-form expression is $P_k = (D^{-1}A)^k$.*

**Definition 2.5** (Simple Path Matrix). *Similarly, we refer to the k-hop simple path matrix $S_k \in \mathbb{N}^{|\mathcal{V}| \times |\mathcal{V}|}$ as the matrix where the $(i, j)$-th entry, $(S_k)_{ij}$, is the number of simple paths of length $k$ from node $i$ to node $j$.*

Unlike $P_k$, $S_k$ does not have a closed-form solution, and computing it is computationally expensive (Vassilevska & Williams, 2009). In the following, we focus on the properties of pairs of nodes $(i, j) \in \mathcal{V} \times \mathcal{V}$ through different *edge encoding* methodologies. To facilitate the comparison with the encoding of another pair of nodes $(i', j')$ in a second graph $\mathcal{G}'$, we introduce the following equivalence relation for $k \geq 1$:

$$(i, j)^{\mathcal{G}} \overset{k}{=}_{\text{RW}} (i', j')^{\mathcal{G}'} \iff (P_k)_{ij} = (P'_k)_{i'j'},$$

where $P'_k$ is the $k$-hop random walk matrix of graph $\mathcal{G}'$. Note that this does not require $\mathcal{G}'$ to have the same number of nodes as $\mathcal{G}$. This can be generalized as:

$$(i, j)^{\mathcal{G}} =_{\text{RW}} (i', j')^{\mathcal{G}'} \iff \forall k \in \mathbb{N}^*, (i, j)^{\mathcal{G}} \overset{k}{=}_{\text{RW}} (i', j')^{\mathcal{G}'}.$$

The same equivalence relations can be defined for simple paths (writing $\overset{k}{=}_{\text{SP}}$ and $=_{\text{SP}}$) by substituting $P_k$ and $P'_k$ with $S_k$ and $S'_k$.

### 2.2. RRWP encoding in Graph Transformers

Pure graph transformers, which do not use any MPNN layer, typically encode structural and positional information directly in the self-attention layer. To compute the

---

[1]The Python implementation of the algorithm is available on the project's Github page.

RRWP matrix $E_{\mathrm{RW}}$, all $k$-hop random walk matrices up to a maximum walk length $K$ are concatenated into a matrix $P = [P_1, \cdots, P_K] \in [0,1]^{|\mathcal{V}| \times |\mathcal{V}| \times K}$, which is then encoded through a shallow neural network $\phi_0$ that maps $K$ to a (usually larger) dimension $d$ (Menegaux et al., 2023; Ma et al., 2023): $E_{\mathrm{RW}} = \phi_0(P) \in \mathbb{R}^{|\mathcal{V}| \times |\mathcal{V}| \times d}$. $E_{\mathrm{RW}}$ both acts as a relative positional encoding, since it contains the shortest path distance between nodes, and as a structural edge encoding, since walks encode information about the visited sub-structures. As a standard positional encoding, it biases the pair-wise alignment between nodes given by the attention matrix. Very generally, the resulting self-attention layer can be written as follows, where the precise implementation of $\phi_1$ and $\phi_2$ varies among different methods:

$$a_{ij} = \phi_1(W^Q x_i, W^K x_j, (E_{\mathrm{RW}})_{ij}) \qquad (1)$$

$$\alpha_{ij} = \frac{a_{ij}}{\sum_k a_{ik}} \qquad (2)$$

$$y_i = \sum_j \alpha_{ij} \phi_2(W^V x_j, (E_{\mathrm{RW}})_{ij}) \qquad (3)$$

with $x_i$ and $x_j$ the features of nodes $i$ and $j$, $W^Q$, $W^K$ and $W^V$ the query, key and value matrices, and $y_i$ the output from the self-attention layer for node $i$.

## 3. Theoretical Properties of RRWP and SPSE

In this section, we highlight the limitations of RRWP in distinguishing different graph structures, as it can assign identical transition probabilities to edges in distinct topologies, potentially leading to suboptimal performance. Conversely, we show that SPSE naturally captures cycle-related information. In particular, SPSE enables cycle counting, which is crucial in applications such as molecular chemistry (e.g., identifying functional groups like aromatic rings) (May & Steinbeck, 2014; Agúndez et al., 2023), social network analysis (e.g., detecting communities) (Radicchi et al., 2004; Dhilber & Bhavani, 2020), and circuit design (e.g., analyzing feedback loops) (Horowitz et al., 1989).

### 3.1. Limitations of RRWP for Edge Encoding

Random walk probabilities possess the great advantage of being computable in closed form. However, their use introduces certain ambiguities, as illustrated in Figure 1. In these two examples, $\mathcal{G}_A$ and $\mathcal{G}_C$ are cycle graphs, $\mathcal{G}_B$ and $\mathcal{G}_D$ are path graphs, and yet the following two relations hold (see Appendix D for a graphical proof up to a depth of 5):

$$(0,1)^{\mathcal{G}_A} =_{\mathrm{RW}} (0,1)^{\mathcal{G}_B},$$
$$(0,1)^{\mathcal{G}_C} =_{\mathrm{RW}} (0,1)^{\mathcal{G}_D}.$$

Although the edges belong to clearly different graphs, RRWP assigns them the same transition probabilities in the cycle and path graphs. These examples are two special

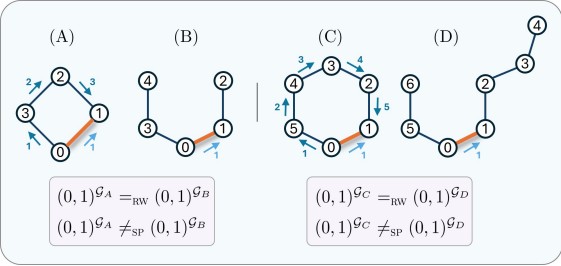

*Figure 1.* RRWP encodes identically edges from very distinct graphs which are separated by SPSE (see the tree decompositions in Appendix D)

cases of the following result which links RRWP edge encodings in even-length cycle graphs and linear graphs (all proofs can be found in Appendix C):

**Proposition 1.** *Let $\mathcal{G} = (\mathcal{V}, \mathcal{E})$ be an even-length cycle graph, i.e. $|\mathcal{V}| = 2n$ for some $n \in \mathbb{N}^*$, and $\mathcal{G}' = (\mathcal{V}', \mathcal{E}')$ a path graph such that $|\mathcal{V}'| = 2n + 1$. Then given any pair of nodes $(i, j)$ in $\mathcal{G}$, there exists a pair of nodes $(i', j')$ in $\mathcal{G}'$ such that $(i, j)^{\mathcal{G}} =_{\mathrm{RW}} (i', j')^{\mathcal{G}'}$.*

Each pair of nodes in an even-length cycle graph is thus equivalent, under RRWP encoding, to another pair of nodes in a linear graph (note that $i$ and $j$ need not be adjacent). In other words, random walks transition probabilities cannot be used to distinguish between even-length cycles and paths when considering single node pairs. On the other end, it is easy to show that this does not apply to SPSE encoding (see Figure 1). This result illustrates how random walk-based structural encoding may fail to capture critical structural differences between node pairs, hence possibly leading to suboptimal performances of the overall graph transformer model.

It is also possible to prove a more general result about the ambiguities of RRWP by leveraging the fact that random walks measure probabilities rather than absolute counts. In particular, the following proposition can be easily proven by induction by considering central symmetries around destination node $j$.

**Proposition 2.** *Let $\mathcal{G} = (\mathcal{V}, \mathcal{E})$ be a graph and $(i, j) \in \mathcal{V} \times \mathcal{V}$ a pair of nodes in $\mathcal{G}$. Then there exists a non-isomorphic graph $\mathcal{G}' = (\mathcal{V}', \mathcal{E}')$ and a pair of nodes $(i', j') \in \mathcal{V}' \times \mathcal{V}'$ such that $(i, j)^{\mathcal{G}} =_{\mathrm{RW}} (i', j')^{\mathcal{G}'}$, i.e. $(i, j)$ and $(i', j')$ are equivalent under RRWP encoding.*

The RRWP encoding of a pair of nodes is therefore never unique, and cannot be used to identify a graph. Of course this is also obviously true for SPSE encoding. However, proving this result for adjacent nodes, in the cases of simple paths, still requires preserving the set of paths that connect them. This hints at the fact that the information contained in simple path edge encodings may be used to distinguish

certain local graph structures, as we discuss in the next section.

### 3.2. SPSE through the Prism of Cycle Counting

The relation between path and cycle counting has been well studied (Perepechko & Voropaev, 2009; Graziani et al., 2024). The following result, which was introduced by Perepechko & Voropaev (2009), connects SPSE encoding with cycle counting for adjacent nodes:

**Proposition 3.** *Let $(i, j)$ be two adjacent nodes in a graph $\mathcal{G} = (\mathcal{V}, \mathcal{E})$, i.e. $(i, j) \in \mathcal{E}$, and $S_k$ the $k$-hop simple path matrix of $\mathcal{G}$ for any $k \in \mathbb{N}^*$, such that $(S_k)_{ij} = m_k \in \mathbb{N}$. Then for $k \geq 2$, there are exactly $m_k$ cycles of length $k + 1$ in $\mathcal{G}$ that admit $(i, j)$ as an edge.*

The case of $k = 1$ simply corresponds to the number of parallel edges between nodes $i$ and $j$. Note that this is not the same as giving the number of cycles that possess $i$ and $j$ as vertices. Two illustrations of this result are presented in Figure 2. Here, $(S_1)_{01} = (S_5)_{01} = 1$ unambiguously defines the edge $(0, 1)$ as belonging to a six-atom cycle. In a second example, SPSE uniquely encodes the double carbon-oxygen bond of any carboxylic acid group. Note that Proposition 3 however does not hold if $(i, j) \notin \mathcal{E}$, and that no equivalent result exists for RRWP since the landing probabilities can be made arbitrarily low by the addition of new edges to a cycle's nodes.

This analysis of structural encodings through the lens of cycles therefore showcases structures for which SPSE is provably more informative than RRWP encoding.

Finally, while the results presented here focus on edge representations, we refer to Appendix B for a discussion on the expressivity of SPSE regarding graph isomorphism.

*Figure 2.* SPSE edge encoding of adjacent nodes characterizes the cycles to which the edge belongs. Thus bonds in the 6-atom cycles of (A) and (B) are encoded identically (they both belong to a single such cycle), and so are the (C=O) bonds of (C) and (D).

**Algorithm 1** Count paths between all pairs of nodes (simplified)

---

1: **Parameters:** Proportion of root nodes $R$, maximum length $K$, maximum DFS depth $D_{\text{DFS}}$, maximum trial number $N$
2: **Input:** Undirected graph $\mathcal{G} = (\mathcal{V}, \mathcal{E})$
3: **Output:** Path count matrix $M \in \mathbb{N}^{|\mathcal{V}| \times |\mathcal{V}| \times K}$
4: $M \leftarrow \mathbf{0}_{|\mathcal{V}| \times |\mathcal{V}| \times K}$ {*Initialize count matrix*}
5: NODES $\leftarrow$ DRAWNODES($R, \mathcal{V}$) {*Select $R \times |\mathcal{V}|$ nodes from $\mathcal{V}$*}
6: **for** each $v$ **in** NODES **do**
7:     DAGS $\leftarrow$ DAGDECOMPOSE($\mathcal{G}$, $v$, $D_{\text{DFS}}$, $N$) {*Retrieve list of node permutations starting with $v$*}
8:     **for** each DAG **in** DAGS **do**
9:         $M \leftarrow$ UPDATE($M$, DAG) {*Update total path count*}
10:     **end for**
11: **end for**
12: **Return:** $M$

---

## 4. Simple Path Structural Encoding

It has been shown that the composition of MPNN layers on paths is more expressive than the 1-WL test (Michel et al., 2023; Graziani et al., 2024). However, due to the combinatorial complexity of the problem, enumerating all paths beyond short path lengths becomes impractical. Results from Section 3 indicate that counting distinct paths between nodes, while requiring significantly less memory, still possesses theoretical advantages over random walk probabilities as an edge structural encoding method. This approach, however, introduces two challenges. First, while existing path-counting algorithms efficiently handle short paths (Perepechko & Voropaev, 2009; Giscard et al., 2019), certain graph topologies and path lengths necessitate approximate methods. Second, since the number of paths between two nodes can grow exponentially (bounded by $\frac{(|\mathcal{V}|-2)!}{(|\mathcal{V}|-k-1)!}$ for length-$k$ paths in a complete graph), an appropriate encoding function is required. This section addresses both challenges.

### 4.1. Simple Path Counting

The SPSE encoding can accommodate any exact or approximate path counting method, although the former are usually restricted to sparse graphs or short path lengths. On the other hand, approximate path counting methods are required when the density of the considered graphs prevents the use of exact methods.

The approximate method followed here relies on the extraction of node orderings from an input undirected graph. Any such ordering can be used to turn the graph into a DAG, which allows to count paths by computing powers of the re-

**Algorithm 2** DAGDECOMPOSE: Decomposition of an input graph into multiple DAGs

---

**Parameters:** Maximum DFS depth $D_{\text{DFS}}$, maximum trial number $N$
**Input:** Graph $\mathcal{G} = (\mathcal{V}, \mathcal{E})$, root node $r$, diameter $D_{\text{MAX}}$
**Output:** List of node orderings $\Pi$
Initialize $\Pi \leftarrow$ EmptyList
**for** $d_{\text{DFS}} = 0$ **to** $D_{\text{DFS}}$ **do**
  **for** $n = 1$ **to** $N$ **do**
    Initialize $\pi \leftarrow$ EmptyList
    **while** $\pi \neq \mathcal{V}$ **do**
      **for** $d = 1$ **to** $D_{\text{MAX}}$ **do**
        **if** $d < d_{\text{DFS}}$ **then**
          $\pi \leftarrow$ DFS$(r, \mathcal{G}, \pi)$ {*Start with $d_{\text{DFS}} - 1$ DFS steps*}
        **else if** $d = d_{\text{DFS}}$ **then**
          $\pi \leftarrow$ PARTIALBFS$(r, \mathcal{G}, \pi)$
        **else**
          $\pi \leftarrow$ BFS$(r, \mathcal{G}, \pi)$ {*Continue with as many BFS steps as possible*}
        **end if**
      **end for**
    **end while**
    $\Pi \leftarrow$ ADD$(\Pi, \pi)$ {*Append $\pi$ to $\Pi$*}
  **end for**
**end for**
**Return:** $\Pi$

---

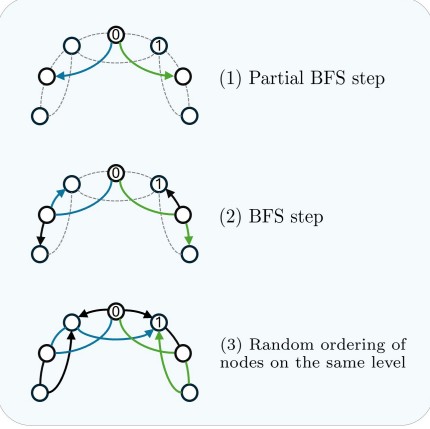

*Figure 3.* Discovering the two paths of length 3 between nodes 0 and 1 in this Circular Skip Link graph requires a partial BFS step in either 0 or 1 (not all child nodes are explored), followed by a BFS step. Nodes that are discovered concurrently are given an arbitrary order, allowing to travel back in the graph.

(1) Partial BFS step

(2) BFS step

(3) Random ordering of nodes on the same level

sulting adjacency matrix. For paths of length $K$, this process incurs a computational complexity of $\mathcal{O}(K|\mathcal{V}|^3)$. Different DAG decompositions of the input undirected graph allow the exploration of different paths, leading to increasingly more accurate path counts. We summarize the overall procedure in Algorithm 1. The detailed version of this algorithm can be found in Appendix E. It consists in updating running path counts for all pairs of nodes and path length $k$, by comparing the counts yielded by each DAG with the stored values, and then storing the maximum of the two. The DAG mining function is presented in Algorithm 2. Starting from a root node, it decomposes a graph into a tree by combining DFS and BFS: a DFS search is initiated at a given root node, updating a list which is incremented with each newly visited node. After it reaches a distance $D_{\text{DFS}}$ from the root node, the search switches to BFS until it cannot proceed further, and the whole process is repeated until all nodes are visited. A DAG can be obtained from the node ordering by directing edges towards nodes of higher indices. DFS allows long path discoveries but cannot simultaneously discover more than one path between nodes, while BFS alone will typically miss long paths, hence their use in combination. We add a *partial BFS* step between the two, which randomly explores a subset of the child nodes, thus allowing to travel through otherwise inaccessible paths (see Figure 3 for an illustra-

tion). Additional parameters include a repetition number $N$ which accounts for random effects, such as the direction taken by the DFS, and a parameter $R$ controlling the proportion of root nodes. The computational cost is dominated by the powers of the adjacency matrix which must be computed for each DAG, for a maximal complexity of $\mathcal{O}(KRD_{\text{DFS}}N|\mathcal{V}|^3)$. The additional factor $RD_{\text{DFS}}N$ can be of the order of tens to hundreds depending on the input graph, which makes SPSE calculation significantly more expensive than RRWP. This needs however to be computed only once as a pre-processing step, and it is also much less than the total number of possible DAG decompositions $2^{|\mathcal{E}|}$, highlighting the effectiveness of the tree decomposition approach. We discuss the choice of hyperparameters and their importance regarding path counts in Section 5.3 and provide actual values in Appendix A.

### 4.2. Path Count Encoding

Compositions of logarithm functions are used to map the obtained path count matrix $S = [S_1, \cdots, S_K]$ to a manageable value range for subsequent neural networks, as total counts can grow very large. Using superscript to denote the composition of a function with itself, we use the following mapping $f$ for a total count $x$:

$$f : x \mapsto \alpha g^n(x) + \beta, \tag{4}$$

with $g : x \mapsto \ln(1 + x)$, and $\alpha$, $\beta$ and $n$ being hyperparameters to be adjusted for different graph collections. Normalized path counts $f(S)$ can then be used in place of the random walk matrix $P$ as input to the edge encoding network of graph transformer models, yielding the SPSE matrix $E_{\text{SP}}$ which replaces $E_{\text{RW}}$ in equations 1 and 3.

# 5. Experiments

We first validate Proposition 3 experimentally through a synthetic experiment (Section 5.1), and demonstrate the empirical superiority of SPSE on real-world datasets (Section 5.2). We then present an ablation study on the algorithm parameters (Section 5.3) and discuss limitations (Section 5.4).

## 5.1. Cycle Counting Synthetic Experiment

**Synthetic Dataset.** To validate Proposition 3, we design a synthetic dataset consisting of 12,000 graphs. Each graph is generated by randomly adding cycles of lengths between 3 and 8 until the total count for each cycle length reaches a value between 0 and 14. This process results in graphs with an average of 149 nodes and 190 edges. The dataset is split into training (10,000 graphs), validation (1,000 graphs), and test (1,000 graphs) sets. The objective is to determine the number of cycles for each of the six cycle lengths. We frame this as six simultaneous multiclass classification tasks and evaluate performance using mean accuracy. Examples of the generated graphs are provided in Appendix F.

**Models.** We build upon two state-of-the-art graph transformer models that use RRWP as an edge encoding method: GRIT (Ma et al., 2023) and CSA (Menegaux et al., 2023). SPSE can seamlessly replace RRWP in these models by substituting the encoding matrix $E_{\text{SP}}$, which captures path counts, for $E_{\text{RW}}$ in equations 1 and 3. To evaluate performance on the cycle counting task, we train these models using three hyperparameter configurations adopted from (Menegaux et al., 2023). These correspond to the setups used for ZINC (config #1), PATTERN (config #2), and CIFAR10 (config #3), covering a range of model complexities from 40 to 280 gigaflops. This provides a comprehensive assessment of the impact of the two edge encoding methods across diverse settings.

**Results.** The test accuracy for both model architectures across the three training configurations is reported in Figure 4, along with standard deviations computed over 10 runs with different random seeds.

In all but one case, SPSE encoding achieves significantly higher cycle counting accuracy than RRWP. All models learn to count cycles almost perfectly under the third training configuration, which suggests that deep architectures can compensate for expressivity limitations in the edge encoding matrix (see Appendix F for details on configurations). However, SPSE still performs significantly better than RRWP for CSA in this setting.

These results empirically validate the superior ability of simple paths to characterize cycles when used as an edge encoding method in graph transformers.

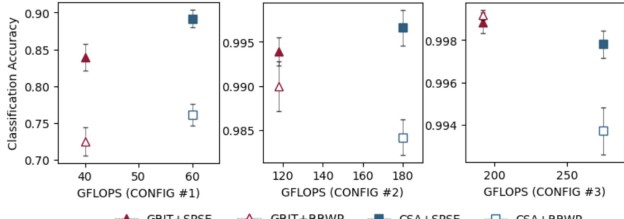

*Figure 4.* Cycle counting accuracies for three training configurations of CSA and GRIT with either RRWP or SPSE edge encoding.

## 5.2. Real-World Benchmarks

**Datasets.** We conduct experiments on graph datasets from three distinct benchmarks, covering both node- and graph-level tasks. These include ZINC, CLUSTER, PATTERN, MNIST, and CIFAR10 from Benchmarking GNNs (Dwivedi et al., 2023), Peptides-functional and Peptides-structural from the Long-Range Graph Benchmark (Dwivedi et al., 2022), and the 3.7M-sample PCQM4Mv2 dataset from the Large-scale Graph Regression Benchmark (Hu et al., 2021). We shall see in Section 5.3 how constraints imposed by graph complexity and dataset sizes impact the path count precision in molecular datasets (ZINC, the two Peptides & PCQM4Mv2), image superpixel (MNIST & CIFAR10) and Stochastic Block Model (SBM) (PATTERN & CLUSTER) benchmarks, justifying a differenciated treatment.

**Experimental Setup.** As before we replace RRWP with SPSE in GRIT and CSA. We also explore adding SPSE to the well-known **GraphGPS** model (Rampášek et al., 2022), in which case the edge encoding is restricted to the elements of $\mathcal{E}$, and is only used to produce node-level positional encodings in the MPNN layer. In all cases, for better result robustness, *we retrain the original model and the SPSE version on ten random seeds* (one seed for the large PCQM4Mv2) using the released training configurations, with two exceptions: the unstable CSA learning rate for ZINC is reduced by a factor 2, and configuration files are added to train CSA on Peptides.

It is important to emphasize that *no hyperparameter tuning* is performed. This decision ensures a fair and unbiased comparison, isolating the contribution of SPSE as a drop-in replacement for RRWP, and demonstrating its effectiveness across different architectures without the need for task-specific adjustments.

Note that replacing walks by path count is done at *no additional cost* as the number of trainable parameters remains unchanged. Two-sided Student's t-tests are conducted to assess the significance of the obtained results. Finally, we compare with the following GNN methods: GCN (Kipf & Welling, 2017), GIN (Xu et al., 2019), GAT (Veličković

*Table 1.* Test results on all eight benchmarks. GPS, CSA & GRIT w/ and w/o SPSE were re-trained on 10 random seeds (**hence variations from seminal works**) on all datasets (one run for PCQM4Mv2). Underline indicates a difference between a baseline and the SPSE model version, and * is used for significant gaps based on a two-sided t-test ($p$-value $\leq 0.05$). Highlighted are best, second, and third best scores. †GPS+GPSE was not retrained and is therefore treated separately.

| Model | Molecular | | | | SBM | | Superpixel | |
|---|---|---|---|---|---|---|---|---|
| | ZINC | Peptides-func | Peptides-struct | PCQM4Mv2 | PATTERN | CLUSTER | MNIST | CIFAR10 |
| | MAE ↓ | AP ↑ | MAE ↓ | MAE ↓ | Accuracy ↑ | Accuracy ↑ | Accuracy ↑ | Accuracy ↑ |
| GCN | 0.367 ± 0.011 | – | – | 0.1379 | 71.892 ± 0.334 | 68.498 ± 0.976 | 90.705 ± 0.218 | 55.710 ± 0.381 |
| GIN | 0.526 ± 0.051 | – | – | 0.1195 | 85.387 ± 0.136 | 64.716 ± 1.553 | 96.485 ± 0.252 | 55.255 ± 1.527 |
| GAT | 0.384 ± 0.007 | – | – | – | 78.271 ± 0.186 | 70.587 ± 0.447 | 95.535 ± 0.205 | 64.223 ± 0.455 |
| GINE | – | 0.5498 ± 0.0079 | 0.3547 ± 0.0045 | – | – | – | – | – |
| GatedGCN | 0.282 ± 0.015 | – | – | – | 85.568 ± 0.088 | 73.840 ± 0.326 | 97.340 ± 0.143 | 67.312 ± 0.311 |
| GIN-AK+ | 0.080 ± 0.001 | – | – | – | 86.850 ± 0.057 | – | – | 72.190 ± 0.130 |
| SAN (+RRWP) | 0.139 ± 0.006 | 0.6439 ± 0.0075 | 0.2545 ± 0.0012 | – | 86.581 ± 0.037 | 76.691 ± 0.65 | – | – |
| Graphormer | 0.122 ± 0.006 | – | – | 0.0864 | – | – | – | – |
| EGT | 0.108 ± 0.009 | – | – | – | 86.821 ± 0.020 | 79.232 ± 0.348 | 98.173 ± 0.087 | 68.702 ± 0.409 |
| GraphViT | 0.073 ± 0.001 | 0.6970 ± 0.0080 | 0.2475 ± 0.0015 | – | – | – | 97.422 ± 0.110 | 73.961 ± 0.330 |
| Exphormer | – | 0.6527 ± 0.0043 | 0.2481 ± 0.0007 | – | 86.742 ± 0.015 | 78.071 ± 0.037 | 98.550 ± 0.039 | 74.696 ± 0.125 |
| Drew | – | 0.7150 ± 0.0044 | 0.2536 ± 0.0015 | – | – | – | – | – |
| GPS + GPSE† | 0.065 ± 0.003 | 0.6688 ± 0.0151 | 0.2464 ± 0.0025 | – | – | – | 98.08 ± 0.13 | 72.31 ± 0.25 |
| *SPSE used in MPNN for node representation* | | | | | | | | |
| GPS | 0.070±0.003 | 0.6601 ± 0.0061 | 0.2509 ± 0.0017 | 0.0942 | 86.688±0.073 | 77.969±0.161 | 98.064±0.157 | 72.097±0.475 |
| **GPS+ SPSE** *(ours)* | 0.068±0.003 | 0.6608 ± 0.0063 | 0.2506 ± 0.0010 | 0.0934 | 86.834±0.025* | 78.440±0.177* | 98.105±0.158 | 72.114±0.462 |
| *RW/SPSE used as edge feature in self-attention layers* | | | | | | | | |
| CSA-RRWP | 0.069±0.003 | 0.6513±0.0061 | 0.2486 ± 0.0012 | 0.0918 | 87.008±0.062 | 79.071±0.120 | 98.127±0.123 | 73.885±0.348 |
| **CSA-SPSE** *(ours)* | 0.061±0.003* | 0.6605 ± 0.0096* | 0.2482 ± 0.0019 | 0.0911 | 87.064±0.052* | 78.940±0.132 | 98.269±0.078* | 73.897±0.524 |
| **GRIT-RRWP** | 0.065±0.005 | 0.6803±0.0085 | 0.2480 ± 0.0025 | 0.0838 | 87.229±0.056 | 79.730±0.189 | 98.231±0.197 | 76.246±0.954 |
| **GRIT-SPSE** *(ours)* | 0.059±0.001* | 0.6945±0.0113* | 0.2449±0.0018* | 0.0831 | 87.235±0.040 | 79.571±0.122 | 98.294±0.147 | 77.022±0.430* |

et al., 2018), GatedGCN (Bresson & Laurent, 2017) and GIN-AK+ (Zhao et al., 2022), along with the graph transformers: SAN (Kreuzer et al., 2021), Graphormer (Ying et al., 2021), EGT (Hussain et al., 2022), GraphViT (He et al., 2023), Exphormer (Shirzad et al., 2023), Drew (Gutteridge et al., 2023) and GPSE (Cantürk et al., 2024). Results are reported in Table 1.

**Results and Discussion** Replacing RRWP with SPSE improves performance in 21 out of 24 cases (underlined in Table 1), with variations depending on the benchmark and model. The most notable gains are observed for CSA and GRIT on molecular graphs, where SPSE achieves statistically significant improvements in 5 out of 6 cases (marked with "*"). Improvements are also evident on superpixel benchmarks, with two cases showing statistically significant differences, on both CSA and GRIT. These results indicate that leveraging the full structural encoding matrix in self-attention layers, rather than restricting it to $\mathcal{E}$ as done in GPS, is an effective strategy for enhancing performance. The limited improvement for GPS is however also likely due to the constraint of leaving the total number of parameters unchanged. In practice, this amounts to reducing the dimensionality of existing edge embeddings when using SPSE, limiting its potential benefit. This explanation is further supported by the significant improvements observed on SBM benchmarks, where GPS does not incorporate any

edge encoding. In contrast, pure graph transformers do not benefit from SPSE on CLUSTER. A possible explanation is the increased difficulty of accurately counting paths in the densely connected SBM graphs, leading to underestimated counts. Further details on these limitations are provided in Section 5.4.

### 5.3. Path Count Sensitivity to Hyperparameters

In this section, we examine how the number of paths discovered by Algorithm 1 is affected by variations in hyperparameters across different benchmarks. Those parameters are the proportion of root nodes $R$, the maximum DFS depth $D_{\text{DFS}}$ and the number of trials $N$ (path length $K$ is fixed to walk lengths). ZINC, PATTERN, and MNIST are used as representative benchmarks for molecular graphs, SBM, and superpixels, respectively. We measure the average proportion of discovered paths relative to a canonical configuration (reported in Appendix A) and the computation time per sample while varying a single hyperparameter. Results are reported in Figure 5, with path count proportions as solid lines (left y-axis) and computation time as dashed lines (right y-axis). The main parameter controlling the path count precision for ZINC and other molecular datasets is the root node proportion $R$ which can be set to its maximum value ($R = 1$) due to its low computational cost. In contrast, DFS depth and the number of trials have little to no impact

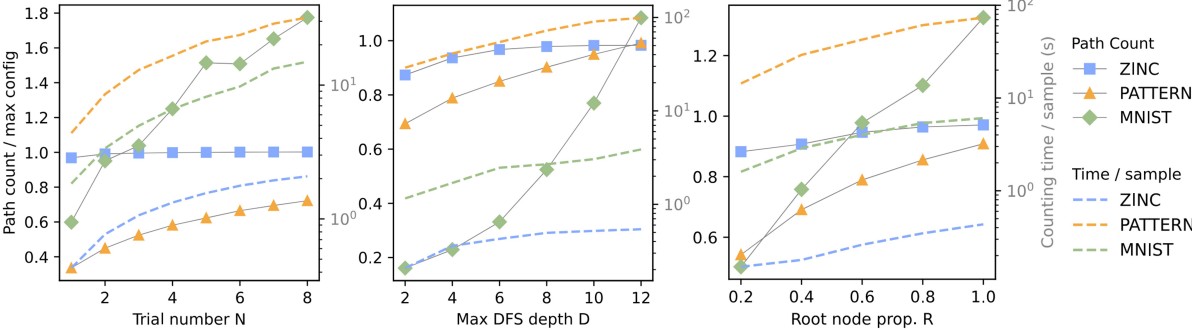

*Figure 5.* Proportion of discovered paths and computation time per sample when varying one hyperparameter at a time, across ZINC, PATTERN, and MNIST datasets. Solid lines (left y-axis) represent the proportion of discovered paths, while dashed lines (right y-axis) indicate computation time per graph sample.

on path counts and can be kept at moderate or low values. MNIST benefits almost equally from increases in all three hyperparameters, though adjusting $D_{\text{DFS}}$ offers the lowest computational overhead. Notably, the trade-off setting of $N = 2$, imposed by the large size of the dataset, results in approximately $40\%$ fewer discovered paths compared to $N = 8$, potentially leading to suboptimal learning accuracy. For the PATTERN dataset, we prioritize increasing $N$ over other hyperparameters, as it provides the most computationally efficient way to improve path discovery. In this case, however, we expect a non-negligible fraction of paths to be missed due to the highly connected nature of SBM graphs, which sets a limit on the number of discoverable paths (see Section 5.4 for further discussion).

### 5.4. Limitations

In the proposed algorithm, the path count between two nodes is updated by comparing the value provided by each new DAG with the total currently stored in memory. The latter is then replaced by the maximum of the two values. This approach is necessary since paths are not enumerated because of memory constraints. The obtained counts therefore constitute lower bounds on the exact number of paths (see line 16 of Algorithm 3 in Appendix E). Failure cases can however arise because individual paths are not distinguished. For instance, consider the input graph illustrated in Figure 6. The total number of paths of length 4 between nodes 0 and 1 is two (blue and green). The only directed graphs that would allow to discover these paths simultaneously are not acyclic: our algorithm can simply count them one at a time.

This limitation is especially true in high-density graphs, such as the CLUSTER dataset where our method does not yield any significant improvement. This suggests that there may exist cases where inaccurate path counts are detrimental to the overall performance, and where it might be preferable to trade longer path lengths for exact counts on shorter paths.

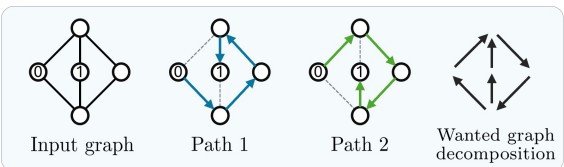

*Figure 6.* Discovering simultaneously the two paths of length 4 between nodes 0 and 1 requires the directed graph decomposition on the right, which violates the acyclic property.

## 6. Related Work

**Graph Transformers.** Graph transformers are a class of models designed to process graph-structured data by leveraging self-attention, which provides a graph-wide receptive field (Vaswani, 2017; Ying et al., 2021), thereby avoiding common limitations of message-passing GNNs (Topping et al., 2022). Despite their advantages, graph transformers have not yet become the dominant architecture in graph learning, primarily due to the lack of unified structural and positional encodings that generalize well across diverse graph tasks (Rampášek et al., 2022). This is evident from the continued success of hybrid approaches that integrate message-passing with global self-attention (Wu et al., 2021; Rampášek et al., 2022; He et al., 2023; Choi et al., 2024), highlighting ongoing opportunities for architectural improvements. Promising research directions include: more flexible spectral attention mechanisms (Bo et al., 2023), models that mitigate the quadratic memory footprint of self-attention (Shirzad et al., 2023), and hierarchical encodings that enhance structural representations (Luo et al., 2024a).

**Expressivity of Path-Based MPNNs.** The expressivity of message-passing neural networks (MPNNs) on paths has been studied in relation to the WL isomorphism test (Michel et al., 2023; Graziani et al., 2024). Notably, Graziani et al. (2024) demonstrated that iteratively updating node features via message-passing along paths originating from these nodes is strictly more expressive than the 1-WL test. How-

ever, these works consider only short path lengths, limiting their applicability to more complex graph structures.

**Counting Paths and Cycles in Graphs.** The problem of counting paths and cycles in graphs has been extensively studied over the past decades (Johnson, 1975; Alon et al., 1997; Flum & Grohe, 2004). Perepechko & Voropaev (2009) derived an explicit formula for counting paths of length up to 6 between two nodes, while Giscard et al. (2019) proposed an algorithm for counting paths of any length based on enumerating connected induced subgraphs. However, these methods become impractical for the large graphs and long path lengths considered in our work, necessitating the use of efficient approximate counting methods.

## 7. Conclusion

This work introduced Simple Path Structural Encoding (SPSE), a novel structural encoding method for graph transformers that leverages path counts instead of random walk probabilities. By providing a more structurally informative edge encoding, SPSE improves performance across various graph learning benchmarks. Our theoretical and experimental study shows that SPSE mitigates the limitations of random walk-based encodings while maintaining computational feasibility through an efficient approximation algorithm. While promising, SPSE's applicability to extremely large-scale graphs and its interaction with different transformer architectures would require further exploration. In particular, SPSE bears particularly promising synergies with hierarchical methods, which would allow efficient path mining at higher hierarchy levels while capturing valuable long-range structural information. Future research directions may also include optimizing its computational efficiency and extending its applicability to broader domains such as knowledge graphs and large social networks.

## Acknowledgments

This work was funded in part by a grant from Fondazione Caritro, under project GenIE. AL and AP acknowledge the support of the MUR PNRR project FAIR - Future AI Research (PE00000013) funded by the NextGenerationEU.

## Impact Statement

This paper presents Simple Path Structural Encoding (SPSE), a novel structural encoding method for graph transformers, aimed at enhancing the expressivity of self-attention mechanisms by capturing richer structural patterns than existing random walk-based encodings. By improving edge encodings with simple path counts, SPSE contributes to the broader field of graph representation learning, offering a more robust method for tasks involving molecular graphs, social networks, and long-range dependencies.

The societal impact of this work aligns with the broader advancements in Machine Learning and Graph Neural Networks, with potential applications in drug discovery, recommendation systems, and scientific knowledge extraction. However, as with any machine learning model, the deployment of SPSE-based architectures should be approached with considerations for fairness, interpretability, and robustness, particularly in high-stakes applications.

We do not foresee any immediate ethical concerns or risks associated with our contributions beyond those generally applicable to graph machine learning. Future work should consider potential biases in training data and energy efficiency of graph transformers, as large-scale models can have computationally intensive requirements.

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

*Table 2.* Statistic, path counting hyperparameters and total counting time per dataset with the Python implementation of the algorithm.

| Dataset | ZINC | PATTERN | CLUSTER | MNIST | CIFAR10 | Peptides | PCQM4Mv2 |
|---|---|---|---|---|---|---|---|
| *Dataset Statistics* | | | | | | | |
| # of graphs | 12,000 | 14,000 | 12,000 | 60,000 | 70,000 | 15,535 | 3,746,620 |
| Avg. # of nodes per sample | 23.2 | 118.9 | 117.2 | 70.6 | 117.6 | 150.9 | 14.1 |
| Avg. # of edges per sample | 24.9 | 3,039.3 | 2,150.9 | 564.5 | 941.1 | 307.3 | 14.6 |
| Avg. density | 0.10 | 0.43 | 0.36 | 0.23 | 0.14 | 0.03 | 0.16 |
| *Path count* | | | | | | | |
| $R$ (# of root nodes, % of $|\mathcal{V}|$) | 100% | 40% | 40% | 55% | 55% | 100% | 100% |
| $K$ (max length) | 20 | 16 | 16 | 17 | 17 | 23 | 15 |
| $D_{\text{DFS}}$ (max DFS depth) | 6 | 2 | 2 | 11 | 11 | 4 | 6 |
| $N$ (# of trials per depth) | 1 | 7 | 7 | 2 | 2 | 1 | 1 |
| Time (hr) | 1 | 60 | 39 | 34 | 80 | 48 | 80 |
| *Model training* | | | | | | | |
| $\alpha$ | 0.5 | 0.2 | 0.2 | 0.2 | 0.2 | 0.2 | 0.5 |
| $\beta$ | 0 | -0.2 | -0.2 | -0.2 | -0.2 | -0.2 | 0 |
| $n$ | 1 | 3 | 3 | 3 | 3 | 2 | 1 |

## A. Additional Path Count Statistics and Training Hyperparameters

Typical hyperparameter values for path counts with the respective computing duration are reported in Table 2. The hyperparameter search was conducted on a reduced pool of graphs and root nodes to optimize the trade-off between exhaustivity and duration. The reported figures should therefore not be viewed as optimal, but rather give an indication of how they affect the path count duration. The latter ranges from under one hour for ZINC to a bit more than three days for CIFAR10 and the large PCQM4Mv2 dataset which contains 3.7M graphs. Once computed, path counts are stored and used for model training at no additional cost.

**Path count encoding parameters $\alpha$, $\beta$ and $n$.** In the absence of clear heuristics regarding the effects of parameters $\alpha$, $\beta$ and $n$ of Equation (4), we report in Table 2 the values that led to the best results for each dataset. Among those, $n$ primarily controls the compression of the dynamic range of path counts, while both $\alpha$ and $n$ contribute to the adjustment of the output range of function $f$ in Equation (4). Unsurprisingly, denser datasets call for a higher $n$, a lower $\alpha$, or a combination of both.

## B. Expressivity of SPSE encoding

We provide here a discussion about the expressivity of the proposed SPSE encoding in light of findings from previous works. A reasoning similar to the one followed for Path-WL (Graziani et al., 2024) can be used to show that an iterative node coloring algorithm based on global attention with SPSE is more expressive than 1-WL. We note also that the results regarding the expressivity of GRIT remain valid when SPSE replaces RRWP as the chosen structural encoding method, except for the case of Proposition 3.1 (b) which requires access to transition probabilities.

However, no previous study allows the comparison of SPSE and RRWP expressivity. Contrary to WL- / Path-trees used in Graziani et al. (2024) and Michel et al. (2023), SPSE and RRWP aggregate information over simple paths and walks without enumerating them, which prevents one from using the strategy of the proof of Theorem 3.3 of Michel et al. (2023) to conclude. We leave the task of proving whether both encodings can be compared for future work.

## C. Proofs

### C.1. Proof of Proposition 1

**Proposition.** *Let $\mathcal{G} = (\mathcal{V}, \mathcal{E})$ be an even-length cycle graph, i.e. $|\mathcal{V}| = 2n$ for some $n \in \mathbb{N}^*$, and $\mathcal{G}' = (\mathcal{V}', \mathcal{E}')$ a path graph such that $|\mathcal{V}'| = 2n + 1$. Then given any pair of nodes $(i, j)$ in $\mathcal{G}$, there exists a pair of nodes $(i', j')$ in $\mathcal{G}'$ such that $(i, j)^{\mathcal{G}} =_{\text{RW}} (i', j')^{\mathcal{G}'}$.*

*Proof.* The proof is done by induction by considering landing probabilities of walks of length $k$ and $k + 1$ for any given $k$ in $\mathbb{N}^*$. To do that, we use the following notations:

$$p_k(i, j) = (P_k)_{ij}$$

where $P_k$ is the $k$-hop random walk matrix of $\mathcal{G}$, and $(P_k)_{ij}$ is the random walk probability of length $k$ between nodes $i$ and $j$. We also take advantage of the symmetries of the circular graph $\mathcal{G}$ to denote as $p_k(j)$ the $k$-hop walk probability between any two nodes distant of $j \leq n$. As $\mathcal{G}'$ is a linear graph with an odd number of nodes, we also use the notation

$$p'_k(i, j) = (P'_k)_{ij},$$

with $P'_k$ the $k$-hop random walk matrix of $\mathcal{G}'$, but use this time node indices to represent the distance from the central node of the graph. Hence $p'_k(0, n)$ is the $k$-hop walk probability between the central node and any of the two extremity nodes. Similarly, for all $1 \leq j \leq n$, $p'_k(j)$ refers to $k$-hop walks originating from the central node and ending on one of the two nodes distant of $j$.

The first part of the proof consists in showing that, for all $1 \leq j < n$ and all $k \in \mathbb{N}^*$, $p'_k(j) = p_k(j)$, and $p'_k(n) = \frac{1}{2}p_k(n)$. The initialization, up to $k = j$, is straightforward as all nodes have exactly two neighbors, except from the extremity nodes in $\mathcal{G}'$ which have one. Suppose that the previous statement holds for $k \geq 1$. Then we have:

$$p'_{k+1}(j) = p'_k(j - 1) \times \frac{1}{2} + p'_k(j + 1) \times \frac{1}{2}$$
$$= p_{k+1}(j) \quad \text{if } j < n - 1,$$

$$p'_{k+1}(n - 1) = p'_k(n - 2) \times \frac{1}{2} + p'_k(n)$$
$$= p_k(n - 2) \times \frac{1}{2} + p_k(n) \times \frac{1}{2}$$
$$= p_{k+1}(n - 1),$$

$$p'_{k+1}(n) = p'_k(n - 1) \times \frac{1}{2}$$
$$= (2 \times p_k(n - 1) \times \frac{1}{2}) \times \frac{1}{2}$$
$$= \frac{1}{2}p_{k+1}(n).$$

At this point, equivalent random walk edge encodings have been found in $\mathcal{G}'$ for any pair of nodes of $\mathcal{G}$ distant of less than $n$ from each other. The last part of the proof consists in showing that $p_k(n) = p'_k(n, 0)$ for all $k \geq 1$, or equivalently, from the previous result, that $p'_k(n, 0) = 2p'_k(n)$.

This is again done by induction, by proving the following statement for all $i, j \in \mathbb{N}^2$, $k \in \mathbb{N}^*$ such that $0 \leq i < j \leq n$:

$$p'_k(i, j) = p'_k(j, i) \quad \text{if } j < n,$$
$$p'_k(i, n) = \frac{1}{2}p'_k(n, i).$$

As before, the initialization up to the distance between $i$ and $j$ is direct as all nodes have exactly two neighbors, except $n$ which has one. Suppose now that the statement holds for some $k \leq 1$. Then, if $j < n$:

$$p'_{k+1}(i, j) = p'_1(i, i + 1)p'_k(i + 1, j) + p'_1(i, i - 1)p'_k(i - 1, j)$$
$$= p'_k(j, i + 1)p'_1(i + 1, i) + p'_k(j, i - 1)p'_1(i - 1, i)$$
$$= p'_{k+1}(j, i),$$

and

$$p'_{k+1}(i, n) = p'_k(i, n-1)p'_1(n-1, n)$$
$$= \frac{1}{2} \times p'_1(n, n-1)p'_k(n-1, i)$$
$$= \frac{1}{2} \times p'_{k+1}(n, i),$$

which concludes the proof.

$\square$

### C.2. Proof of Proposition 2

**Proposition.** *Let $\mathcal{G} = (\mathcal{V}, \mathcal{E})$ be a graph and $(i, j) \in \mathcal{V} \times \mathcal{V}$ a pair of nodes in $\mathcal{G}$. Then there exists a non-isomorphic graph $\mathcal{G}' = (\mathcal{V}', \mathcal{E}')$ and a pair of nodes $(i', j') \in \mathcal{V}' \times \mathcal{V}'$ such that $(i, j)^{\mathcal{G}} =_{\mathrm{RW}} (i', j')^{\mathcal{G}'}$, i.e. $(i, j)$ and $(i', j')$ are equivalent under RRWP encoding.*

*Proof.* We proceed by induction to show that given an edge $(i, j)$ in a graph $\mathcal{G}$, there always exists a non-isomorphic graph $\mathcal{G}'$ with an edge $(i', j')$ such that $(i, j)^{\mathcal{G}} =_{\mathrm{RW}} (i', j')^{\mathcal{G}'}$. Without loss of generality, we take $i = i' = 0$ and $j = j' = 1$, and denote probabilities in $\mathcal{G}'$ with a prime superscript. We write $p_{ij} = (P_1)_{ij}$, $p'_{i'j'} = (P'_1)_{i'j'}$ with $P_k$ and $P'_k$ the $k$-hop random walk matrices of $\mathcal{G}$ and $\mathcal{G}'$. Finally, we also write, $\forall k \in \mathbb{N}$, $p_k(j) = (P_k)_{0j}$ and $p'_k(j') = (P'_k)_{0j'}$.

The proof relies on the introduction of a graph $\mathcal{G}'$ as the result of a central symmetry of $\mathcal{G}$ around node 1. Concretely, each node $i$ (respectively $j$) originally in $\mathcal{G}$ possesses a symmetric node $i'$ (respectively $j'$) in $\mathcal{G}'$ such that:

$$\forall i, j \in \mathcal{G}, i \neq 1, \quad p'_{i'j'} = p'_{ij} = p_{ij} \tag{5}$$

$$\forall j \in \mathcal{N}_1 \cap \mathcal{G}, \quad p'_{1j} = p'_{1j'} = \frac{1}{2}p_{1j}, \tag{6}$$

where $\mathcal{N}_1$ is the neighborhood of node 1 in $\mathcal{G}'$, $\mathcal{N}_1 \cap \mathcal{G}$ is its restriction to $\mathcal{G}$, and $1' = 1$.

The initialization is straightforward as, for such $\mathcal{G}'$ and for any $k$ lesser or equal to the shortest path distance between 0 and 1, it holds true that $p'_k(1) = p_k(1)$. Suppose now that for a given $k$, we have:

$$\forall l \leq k, \quad p'_l(1) = p_l(1). \tag{7}$$

Let $p^*_k(i)$ designate the probability to travel from 0 to $i$ in $k$ steps in $\mathcal{G}$ *without passing through* 1, $p'^*_k(i)$ the similar probability in $\mathcal{G}'$, and let $\hat{p}_k(i)$ and $\hat{p}'_k(i)$ be respectively the probabilities to travel to $i$ in $\mathcal{G}$ and $\mathcal{G}'$ *while visiting* 1 *at least once*. It is then possible to write $p'_{k+1}(1)$ as follows:

$$p'_{k+1}(1) = \underbrace{\sum_{i \in \mathcal{N}_1} p'^*_k(i)p'_{i1}}_{\text{① Walks that do not go through 1}} + \underbrace{\sum_{i \in \mathcal{N}_1} \hat{p}'_k(i)p'_{i1}}_{\text{② Walks that travel through 1 at least once}} \tag{8}$$

The first term rewrites simply:

$$\sum_{i \in \mathcal{N}_1} p'^*_k(i)p'_{i1} = \sum_{i \in \mathcal{N}_1 \cap \mathcal{G}} p'^*_k(i)p'_{i1} \tag{9}$$

$$= \sum_{i \in \mathcal{N}_1 \cap \mathcal{G}} p^*_k(i)p_{i1} \tag{10}$$

For the second term, we introduce $\mathcal{W}^{*l}_{i \to j}$ as the set of random walks of length $l$ between nodes $i$ and $j$ that do not go through 1. Thus, an element $w$ of $\mathcal{W}^{*l}_{i \to j}$ is a multiset of $l$ nodes such that $w = (w_1, \ldots, w_l)$ with $w_1 = i$ and $w_l = j$. We may now

rewrite ②, and in particular each $\hat{p}'_k(i)$ in the sum as follows:

$$\hat{p}'_k(i) = \sum_{l \leq k-1} \sum_{j \in \mathcal{N}_1} \sum_{w \in \mathcal{W}^{*k-(l+1)}_{j \to i}} p'_l(1)p'_{1j} \prod_{m>1} p'_{w_{m-1}w_m} \tag{11}$$

$$= 2 \sum_{l \leq k-1} \sum_{j \in \mathcal{N}_1 \cap \mathcal{G}} \sum_{w \in \mathcal{W}^{*k-(l+1)}_{j \to i}} p_l(1)\frac{p_{1j}}{2} \prod_{m>1} p_{w_{m-1}w_m} \tag{12}$$

$$= \hat{p}_k(i), \tag{13}$$

where equations 5, 6 and 7 are used for (11) → (12). Putting everything back together, we have:

$$p'_{k+1}(1) = \sum_{i \in \mathcal{N}_1 \cap \mathcal{G}} p^*_k(i)p_{i1} + \sum_{i \in \mathcal{N}_1 \cap \mathcal{G}} \hat{p}_k(i)p_{i1} \tag{14}$$

$$= p_{k+1}(1), \tag{15}$$

which concludes the proof. □

### C.3. Proof of Proposition 3

**Proposition.** *Let $(i, j)$ be two adjacent nodes in a graph $\mathcal{G} = (\mathcal{V}, \mathcal{E})$, i.e. $(i, j) \in \mathcal{E}$, and $S_k$ the $k$-hop simple path matrix of $\mathcal{G}$ for any $k \in \mathbb{N}^*$, such that $(S_k)_{ij} = m_k \in \mathbb{N}$. Then for $k \geq 2$, there are exactly $m_k$ cycles of length $k + 1$ in $\mathcal{G}$ that admit $(i, j)$ as an edge.*

*Proof.* This can be proved directly by noting that each cycle of length $k + 1$ on which the edge $(i, j)$ lies is a walk of $k + 2$ nodes of the form $i, ...j, i$ where only the first node is repeated. The restriction of the latter to its $k + 1$ first nodes is itself a path of length $k$ between $i$ and $j$, which therefore increments $(S_k)_{ij}$. Conversely, all paths counted in $(S_k)_{ij}$ can be completed by the edge $ji$ to form a cycle of length $k + 1$, which proves the equality.

□

## D. Tree Decompositions

The formal proof of the equivalence of the RRWP encodings in circular and linear graphs is given in Appendix C.1. Here we give a graphical sketch of the proof for the two examples considered in Figure 1. The tree decompositions of the graphs in Figure 1 are presented in Figure 7 for walks starting in node 0 and a maximum depth of 5. As nodes only have one or two neighbors, colors are used to represent the walk probabilities toward child nodes. Edges $(0, 1)$ have the same random walk probabilities in (A) and (B) (Figure 7, top) on one hand, and in (C) and (D) (Figure 7, bottom) on the other hand, despite belonging to very different graphs. It is straightforward to extend these results to any depth, hence verifying that $(0, 1)^A =_{\text{RW}} (0, 1)^B$, and $(0, 1)^C =_{\text{RW}} (0, 1)^D$.

## E. Full Path Counting Algorithm

We present in Algorithm 3 the detailed version of Algorithm 1. Once a list of node permutations is obtained from the DAGDECOMPOSE function, rows and columns of the adjacency matrix are first permuted according to each new node ordering, which then allows to retain only the elements that are above the diagonal (all edges are directed towards nodes of higher index). Subsequently, $k$-hop path counts are easily obtained by computing powers of the resulting nilpotent matrix $\tilde{A}$, after which rows and columns are permuted back to the original ordering. The transpose of the resulting matrix $A_k$ yields the opposite paths and is then added to $A_k$ before updating the running $k$-hop path count matrix $M_k$. If $\mathcal{G}$ is instead directed, this step is skipped and only $A_k$ is used to update $M_k$.

## F. Synthetic Experiment

Figure 8 shows three instances of randomly generated graphs for experiments in Section 5.1. These graphs possess an average of 149 nodes and 190 edges. We also detail the three configurations used for the synthetic experiments in Table 3.

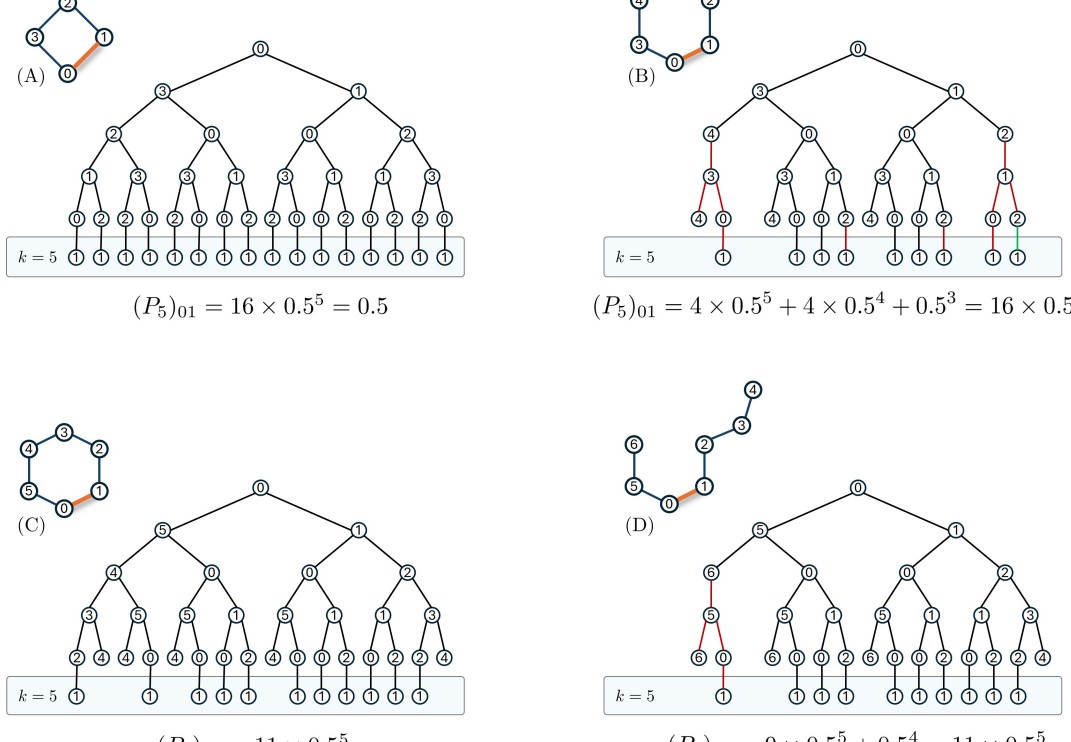

$$(P_5)_{01} = 16 \times 0.5^5 = 0.5$$

$$(P_5)_{01} = 4 \times 0.5^5 + 4 \times 0.5^4 + 0.5^3 = 16 \times 0.5^5$$

$$(P_5)_{01} = 11 \times 0.5^5$$

$$(P_5)_{01} = 9 \times 0.5^5 + 0.5^4 = 11 \times 0.5^5$$

*Figure 7.* Tree decompositions for the graphs in Figure 1 for walks rooted at node 0, up to a maximum depth of 5. Colors denote different probabilities to reach a node at a given depth level $k$: black lines are associated with a probability of $0.5^k$, red lines $0.5^{k-1}$, and green lines $0.5^{k-2}$.

---

**Algorithm 3** Count paths between all pairs of nodes

---

1: **Parameters:** Proportion of root nodes $R$, maximum length $K$, maximum DFS depth $D_{\text{DFS}}$, maximum trial number $N$
2: **Input:** Undirected graph $\mathcal{G} = (\mathcal{V}, \mathcal{E})$, adjacency matrix $A$
3: **Output:** Path count matrix $M \in \mathbb{N}^{|\mathcal{V}| \times |\mathcal{V}| \times K}$
4: $M_1, \ldots, M_K \leftarrow \mathbf{0}_{|\mathcal{V}| \times |\mathcal{V}|}$ {*Initialize count matrices*}
5: **for** $i = 1$ **to** $R \times |\mathcal{V}|$ **do**
6:     $r_i \leftarrow \text{SELECTROOTNODE}(\mathcal{G}, i)$
7:     $\Pi \leftarrow \text{DAGDECOMPOSE}(\mathcal{G}, r_i, D_{\text{DFS}}, N)$ {*Retrieve list of node permutations starting with $r_i$*}
8:     **for** each $\pi_j$ **in** $\Pi$ **do**
9:         $\tilde{A} \leftarrow \text{REORDER}(A, \pi_j)$ {*Permute rows and columns according to $\pi_j$*}
10:         $\tilde{A} \leftarrow \text{TRIUP}(\tilde{A})$ {*Keep elements above the diagonal, fill the rest with $0$*}
11:         $k \leftarrow 1$
12:         $\tilde{A}_k \leftarrow \tilde{A}$
13:         **while** $k \leq K$ **and** $\tilde{A}_k \neq \mathbf{0}_{|\mathcal{V}| \times |\mathcal{V}|}$ **do**
14:             $\tilde{A}_k \leftarrow \tilde{A}_k \times \tilde{A}$
15:             $A_k \leftarrow \text{REORDER}(\tilde{A}_k, \pi_j^{-1})$
16:             $M_k \leftarrow \max(M_k, A_k + A_k^\top)$ {*Add opposite paths and store new maximum counts*}
17:             $k \leftarrow k + 1$
18:         **end while**
19:     **end for**
20: **end for**
21: **Return:** $M_1, \ldots, M_K$

---

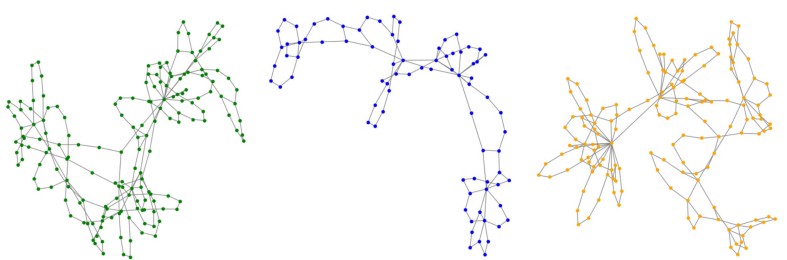

*Figure 8.* Examples of synthetic graphs generated for the cycle counting experiment.

*Table 3.* Model configurations used for the synthetic experiments.

| Configuration | # 1 | # 2 | # 3 |
|---|---|---|---|
| Transformer layers | 3 | 6 | 10 |
| Self-attention heads | 4 | 4 | 8 |
| Hidden dimension | 52 | 64 | 64 |
| Learning rate | $10^{-3}$ | $5 \times 10^{-4}$ | $5 \times 10^{-4}$ |
| Epochs | 100 | 300 | 400 |

