# OpenReview forum: "Simple Path Structural Encoding for Graph Transformers"
_ICML.cc/2025/Conference — ICML 2025 poster_

### Official Review · Reviewer_Luky · 2025-02-28

**Overall Recommendation:** 3

**Summary:**

This paper introduces a novel graph structural encoding method, Simple Path Structural Encoding (SPSE), as a replacement for Random Walk Structural Encoding (RWSE) in graph transformers. SPSE encodes graph structure by counting simple paths of varying lengths between node pairs, providing more accurate structural information compared to random walks. To address the computational challenges of simple path counting, the authors propose an efficient algorithm based on successive DAG decompositions using depth-first search (DFS) and breadth-first search (BFS). This approach avoids the exponential memory costs associated with path enumeration, enabling scalability to longer path lengths.
The key innovations of this paper include:
1.	The introduction of SPSE, a new graph structural encoding method that provides more accurate structural information;
2.	The design of an efficient algorithm to tackle the computational difficulties of simple path counting, demonstrating strong scalability.

**Claims And Evidence:**

The claims presented in the submission are well-supported by clear and compelling evidence. The authors substantiate their arguments with illustrative examples demonstrated in Figure 1 and Figure 2, as well as a detailed comparative analysis presented in Figure 4. Furthermore, the methodology employed for data collection and analysis is robust, thoroughly documented, and enhances the credibility of the findings.

**Essential References Not Discussed:**

There are no essential references missing from the paper.

**Experimental Designs Or Analyses:**

The experimental design incorporates SPSE as an additional component to the existing advanced Graph Transformer architecture, and the implementation results demonstrate a consistent improvement in performance. Alternatively, when SPSE is used to replace the RWSE in these Graph Transformers, the experimental outcomes indicate that SPSE achieves better performance than RWSE.

**Methods And Evaluation Criteria:**

Yes. SPSE mitigates the limitations of random walk-based encodings while maintaining computational feasibility through an efficient approximation algorithm. In addition to its application in this study, the algorithm also serves as an approximate solution to the fundamental graph theory problem known as Simple Path Counting.

**Other Comments Or Suggestions:**

None

**Other Strengths And Weaknesses:**

Although SPSE performs better than RWSE in most cases, the improvement is only slight.

**Questions For Authors:**

None

**Relation To Broader Scientific Literature:**

1.	Random walk structural encoding (RWSE), which encodes richer structural information by considering random walk probabilities as edge features. RWSE has shown substantial improvements in the performance of state-of-the-art graph transformers (“Self-attention in colors: Another take on encoding graph structure in transformers”, TMLR 2023). This paper uses simple path matrix instead of random walk matrix to perform structural encoding.
2.	Simple path counting, existing path-counting algorithms (“A general purpose algorithm for counting simple cycles and simple paths of any length”, Algorithmica) efficiently handle short paths, the graph topologies and path lengths considered necessitate approximate methods.

**Theoretical Claims:**

Yes. I checked the correctness of proofs for Proposition 1, Proposition 2 and Proposition 3.

---

> ### Author Rebuttal · Authors · 2025-03-30
>
> Thank you for your thoughtful and constructive review. We're glad that the clarity of our claims, the robustness of our experimental design, and the correctness of our theoretical contributions were appreciated.
> While no major concern was raised, we would like to contextualize the following point below:
>
> **Q1. Although SPSE performs better than RWSE in most cases, the improvement is only slight.**
>
> **R1.**
> We completely agree that, although the performance gains over RWSE-based methods are statistically significant on 4 out of 7 datasets, they can sometimes appear modest. However, we would like to highlight that these results were obtained only by replacing walk probabilities by path counts, with no further alteration to the model input or architecture.
> That said, we believe there is still untapped potential in SPSE, as suggested in Section 5.3 ("Path Count Sensitivity to Hyperparameters"). Furthermore, additional performance gains may be possible through more extensive tuning of SPSE's three key hyperparameters.

---

### Official Review · Reviewer_r2Mc · 2025-03-05

**Overall Recommendation:** 3

**Summary:**

The work introduces a new perspective on structural encoding for Graph Transformers by leveraging Simple Path Structural Encoding (SPSE). The paper claims to be the first to propose encoding edge information based on simple path counts rather than random walk probabilities, addressing the limitations of Random Walk Structural Encoding (RWSE) in distinguishing local graph structures. It introduces an efficient approximation algorithm for counting simple paths, enabling SPSE to capture richer structural patterns, particularly in graphs with cyclic dependencies. Experimental results demonstrate that SPSE improves performance across various benchmarks, including molecular and long-range graph tasks, outperforming RWSE-based Graph Transformers.

**Claims And Evidence:**

1. In Figure 1, I'm confused why RWSE cannot distinguish A vs. B and C vs. D. Can the author elaborate and provide empirical numbers to prove that? RW of these graphs should be easy and fast enough to compute imo.
- I would like to make a note here that RWSE used in, for example GRIT, is not limited to a single value of k, but it is a concatenation of multiple random walk matrix A_i for i \in {1...k}.

**Essential References Not Discussed:**

N/A

**Experimental Designs Or Analyses:**

1. Some numbers in Table 1 seems to be off. For example, GRIT-RWSE on peptide-struct was reported as 0.2460 in their paper, while the authors provide 0.2480. Can the authors explain?

**Methods And Evaluation Criteria:**

1. For algorithm 2, how can we get the approximated simple path count from the list of node ordering? I believe this is worth mentioning in the paper to clarify the end2end pipeline.

**Other Comments Or Suggestions:**

Please see "Other Strengths And Weaknesses"

**Other Strengths And Weaknesses:**

Generally, the paper offers a new structural encoding method, yet some points need to be addressed as I mentioned in the upper section. I'm willing to raise my score a bit if all of my concerns can be addressed.

**Questions For Authors:**

Please see "Other Strengths And Weaknesses"

**Relation To Broader Scientific Literature:**

The work proposes a new structural encoding method, i.e. simple path counting, to enhance transformers' performance on graph tasks. To the best of my knowledge, the work is the first one to do so.

**Theoretical Claims:**

1. What is the error bound of the proposed approximation algorithm for simple path counting?
2. What is the theoretical reason behind using log() for Equation (4)?

---

> ### Author Rebuttal · Authors · 2025-03-30
>
> We respectfully acknowledge the reviewer’s comments, and hereby attempt to address the different points that were raised.
>
> **Q1. Distinction of graphs A vs. B using RWSE / RWSE is not limited to a single value of k…**
>
> **R1.** We'd be happy to clarify.
> In Figure 1, the objective is not to differentiate between entire graphs (e.g., Graph A vs. Graph B), but rather to compare how individual edges are represented under different encoding methods.
> Taking the concrete example of edge (0,1) in both Graph A and Graph B, the random walk probability of travelling from node 0 to node 1 in $k$ steps is exactly 0.5 if $k$ is odd and 0 if $k$ is even, and *this holds for any* $\textit{k} \in \mathbb{N}^*$.
> This implies that RW-based structural encoding cannot distinguish between edges (0,1) in Graph A and Graph B.
> This property of RWSE is formalized in Proposition 1 of the manuscript, which we prove in Appendix B.1. We also sketch the manual derivation of the random walk probabilities of edge (0,1) of graphs A to D in Figure 7 (Appendix C).
>
> **Q3. How to get the approximated simple path count?**
>
> **R3.** Thank you for pointing this out. The details about the operation done in line 9 of Algorithm 1 (how the running path count is updated for each new node ordering) are given in the full path counting algorithm, in Appendix D: powers of the adjacency matrix given by the new DAG yield new path counts, and running counts are updated by taking the element-wise maximum across all node pairs and path lengths between the new path counts and the current running values. This will be made more explicit in the main text.
>
> **Q4. Error bound of the approximate counting algorithm**
>
> **R4.** Deriving the error bound is an intricate task, which would require identifying the failure cases and estimating their prevalence. It is however possible to empirically quantify this bound by running exact path counts on various graph topologies, within the runtime limits of these algorithms (i.e. largest possible value of maximum path length $K$, shown below for the `NetworkX` implementation).
>
> For ZINC, MNIST and PATTERN, which contain graphs of respective average densities of 0.09, 0.23 and 0.43, we report below the proportion of paths discovered by our approximate path counting method.
>
> | Maximum   Path Length K |         | 2   | 3   | 4   | 5   | 6   | ... | 20  |
> |:---------------------:|:-------:|-----|-----|-----|-----|-----|-----|-----|
> |                       | ZINC    | 100 | 100 | 100 | 100 | 100 | ... |  99 |
> |     Path Count (%)    | MNIST   | 100 |  93 |  79 |  61 | OOM | ... | OOM |
> |                       | PATTERN | 100 | OOM | OOM | OOM | OOM | ... | OOM |
>
> We observe from experiments that trading path count precision for longer path length can improve overall performances, but the precise interplay between exact path count (when they are affordable), maximum path length $K$ and model accuracy would require further study.
>
> **Q5. Reason behind $log$ in Equation (4)**
>
> **R5.** The only constraint on the encoding function of Equation 4 is injectivity. The use of logarithm composition was empirically driven by the need to reduce the very high magnitudes of total path counts reached for several graph collections (up to $10^{13}$ for $k=16$ on CLUSTER).
>
> **Q6. Differences between Table 1 results and original works**
>
> **R6.** Differences between results originally reported by the authors in CSA / GRIT and GPS and the ones in Table 1 are due to the fact that all models were re-trained (using publicly released code and configurations) on 10 random seeds, instead of 4, as is usually done. Retraining was important to compare RWSE and SPSE performances all things being equal, while running experiments on 10 seeds yielded much stronger results. For the large PCQM4Mv2 on which a single experiment was run, we found a MAE of 0.0838 for GRIT-RWSE, improving the reported value of 0.0859: this highlights the need for retraining, even when using the same number of seeds as the original work. The reason for the observation of such differences (be it better or worse) will be made very clear in the final version of the manuscript.

---

> > ### Comment · Reviewer_r2Mc · 2025-04-03
> >
> > Thanks the author for their rebuttal. I have updated my rating.

---

### Official Review · Reviewer_Ybg7 · 2025-03-13

**Overall Recommendation:** 3

**Summary:**

The paper introduces Simple Path Structural Encoding (SPSE), a novel method for encoding graph structures in graph transformers. SPSE leverages simple path counts between node pairs to capture richer structural information compared to traditional random walk structural encoding (RWSE). The authors propose an efficient approximate algorithm for counting simple paths, addressing the computational challenges associated with path enumeration. SPSE is shown to outperform RWSE in various benchmarks, including molecular and long-range graph datasets, demonstrating significant improvements in discriminative tasks. The method is particularly effective in capturing local cyclic patterns, which are crucial in applications like molecular chemistry and social network analysis.

**Claims And Evidence:**

The claims made in the submission are generally supported by clear and convincing evidence. The theoretical analysis highlights the limitations of RWSE in distinguishing between different graph structures, which SPSE overcomes by capturing cycle-related information more effectively. Experimental results on synthetic and real-world datasets validate the superiority of SPSE over RWSE in cycle counting and other graph learning tasks. However, the performance of SPSE on densely connected graphs like the CLUSTER dataset is limited, indicating potential areas for improvement.

**Essential References Not Discussed:**

While the paper cites key works in graph transformers and structural encodings, it might benefit from discussing recent developments in hierarchical encodings and spectral attention mechanisms, which could further enhance SPSE's effectiveness.

**Experimental Designs Or Analyses:**

The experimental designs are sound, involving a range of datasets and model configurations. The use of multiple hyperparameter settings and the retraining of models on different random seeds enhance the robustness of the results. However, the lack of hyperparameter tuning for SPSE might limit its full potential, as noted in the paper.

**Methods And Evaluation Criteria:**

The proposed methods and evaluation criteria are well-suited for the problem at hand. The use of synthetic datasets for cycle counting and real-world benchmarks for molecular and long-range graphs provides a comprehensive assessment of SPSE's effectiveness. The comparison with RWSE and other graph transformer models using standard metrics (e.g., accuracy, mean absolute error) offers a fair evaluation of SPSE's performance.

**Other Comments Or Suggestions:**

Future work could focus on optimizing SPSE's computational efficiency and exploring its applicability to other domains like knowledge graphs.

**Other Strengths And Weaknesses:**

Strengths include the novel approach to edge encoding and the comprehensive experimental validation. Weaknesses include the computational cost of SPSE and its limited performance on densely connected graphs. The paper is well-structured and clear, making it accessible to readers familiar with graph learning concepts.

**Questions For Authors:**

How do you plan to address the computational cost of SPSE for large-scale graphs?

Are there plans to integrate SPSE with other structural encoding methods to further enhance its expressivity?

How might SPSE be adapted for directed graphs or graphs with weighted edges?

**Relation To Broader Scientific Literature:**

SPSE contributes to the broader field of graph representation learning by offering a more expressive edge encoding method. It builds upon recent advances in graph transformers and message-passing neural networks, addressing the need for more effective structural encodings. The work is related to studies on path-based MPNNs and cycle counting algorithms, which have shown the benefits of using paths over random walks for enhancing model expressivity.

**Theoretical Claims:**

The theoretical claims, such as Propositions 1, 2, and 3, are well-supported by proofs provided in the appendices. These propositions highlight the limitations of RWSE and the advantages of SPSE in capturing structural information. However, a detailed review of the proofs would require examining the appendices closely.

---

> ### Author Rebuttal · Authors · 2025-03-30
>
> Thank you for the detailed and constructive review. We appreciate your recognition of SPSE’s novelty, and of the clarity and comprehensiveness of our experimental validation. We're also glad you found the theoretical contributions, algorithmic details, and supplementary material valuable.
> We address the questions below.
>
> **Q1. Addressing the computational cost of SPSE for large-scale graphs**
>
> **R1.** As the computational complexity of path counting incurs a cubic dependency with the number of nodes, scaling to large-scale graphs is a challenge. One possible solution is to apply pooling techniques [1] to reduce the graph size: computing simple paths between communities in a pooled graph would become manageable, and this could be followed by separate path discovery within each community. This shares links with the discussion in the answer to the next question.
> That being said, accurately combining path counts within and between communities presents new and interesting challenges, constituting an avenue for future research.
>
> [1] Bianchi, F. M., & Lachi, V. The expressive power of pooling in graph neural networks. NeurIPS 2023.
>
> **Q2. Plans to combine SPSE with other SEs?**
>
> **R2.**
>  Yes, one particularly promising direction is [2], which is complementary to our work and can also be used as a plug-in module within existing architectures. In particular, we expect that our two offline preprocessing workflows can be efficiently combined: mining paths at higher hierarchy levels can be done more easily because of simpler topologies, while capturing valuable long-range structural information. This will be mentioned in the discussion on future research direction.
>
> [2] Luo et al., Enhancing Graph Transformers with Hierarchical Distance Structural Encoding, NeurIPS 2024
>
>
> **Q3. SPSE for directed/weighted graphs**
>
> **R3.** The only difference that arises in our algorithm when the input graph is directed is that one must account for the fact that the adjacency matrix is no longer symmetric. This case is addressed in the attached Python file by setting the `directed` boolean to True. This prevents the summation with the transposed adjacency matrix in line 16 of the full algorithm (Algorithm 3 in Appendix D).
> In the special case of a DAG input graph, the ``DAGDecompose`` step will be skipped.
> It is also possible to use different edge attribute values (i.e. weights) inside the adjacency matrices, although the interpretation might be more intricate and calls for closer scrutiny.
>
> **Q4. Contextualization with hierarchical encoding / spectral attention**
>
> **R4.** Thank you for the suggestion. Adding a discussion —aligned with our response to **Q2**— on the potential synergies with hierarchical methods, especially, would help strengthen the future research directions section.
> Spectral attention, meanwhile, is natively used within both GRIT and CSA frameworks.
>
>
> **Q5. Further improvement with hyperparameter tuning**
>
> **R5.** We acknowledge that hyperparameter tuning could further improve SPSE's performance. Here, we intentionally chose not to alter the provided configurations for a transparent and fair comparison with existing structural encodings (a procedure that Reviewer wLjL also acknowledged), but future exploration of full SPSE capabilities will likely require adjusting model and optimization parameters.

---

### Official Review · Reviewer_wLjL · 2025-03-14

**Overall Recommendation:** 4

**Summary:**

This work proposes a new structural encoding for empowering graph transformers, SPSE (Simple Path Structure Encoding). Rather than encoding random walk landing probabilities, SPSE encodes simple path counts, demonstrating superior expressivity in scenarios such as cycle counting. The authors propose an algorithm for computing an approximation of simple path counts between nodes show that SPSE outperforms a variety of structural/positional encodings on benchmark datasets.

**Claims And Evidence:**

I am convinced the theoretical claims are correct, and the authors' assessment of SPSE in their experiments are also supported.

**Essential References Not Discussed:**

To my knowledge, the authors discuss all essential references for their research problem.

**Experimental Designs Or Analyses:**

**Strengths**

- The fact that the authors do no hyperparameter tuning on the main results is transparent for fair comparisons with other SEs.
- The use of 10 runs and random seeds with significance testing strengthens the experimental design.

**Weaknesses**

- The datasets used are largely toy datasets. Results would be stronger if some experiments on more "real-world" graphs were used (e.g. MoleculeNet, citation graphs).

**Methods And Evaluation Criteria:**

Using simple path counts is a straightforward and intuitively more powerful approach than random walk landing probabilities, and the proposed method is total sense for this problem. The evaluation metrics and datasets used are all standard in the graph PSE literature.

**Other Comments Or Suggestions:**

Overall, I am convinced by this paper. The advantages of SPSE seem clear, and the experiments convince me of its effectiveness.

Minor typos:

- "there _exits_"

**Other Strengths And Weaknesses:**

**Strengths**

- The authors do a great job of illustrating how their method works and specific failure cases of RWSE that SPSE remedies.
- The limitations section is well-argued, and the failure case of SPSE is easily understood.

**Weaknessees**

- I think a parameter study on $\alpha, \beta,$ and $n$ would be valuable, similar to their study of the effect of $R, N, K, D_dfs$.
- There is no discussion on the WL-expressivity of the proposed encoding. Given that most PSEs have some analysis regarding the WL-hierarchy, this work would be more convincing if the relationship between SPSE and WL were addressed. Similarly, there is no theoretical proof that SPSE is strictly more expressive than RWSE in a certain hierarchy.
- There are some missing SEs in the main benchmark such as [1].

[1] Cantürk et al. Graph Positional and Structural Encoder. ICML 2024.

**Questions For Authors:**

Please address the above weaknesses.

- What aspect of a given graph contributes most to the runtime of the simple path mining algorithm?
- Also, the proposed path count mining algorithm seems to bear some similarity with Node2Vec during the DAG decomposition phase given its combination of BFS and DFS, except the proposed algorithm alternates between phases of explicit DFS/BFS rather than a biased random walk. Can the authors comment on the connection between these works?

**Relation To Broader Scientific Literature:**

This work is related to the graph positional/structural encoding literature, expanding on [1] and [2] to use path counts rather than random walk probabilities to encode structure. It is also following a line of work that uses counts of important topological features within a graph to supplement neural network features. For example, [3] and [4] explore counting specific subgraph structures.

[1] Menegaux et al. Self-Attention in Colors: Another Take on Encoding Graph Structure in Transformers. TMLR.

[2] Ma et al. Graph Inductive Biases in Transformers without Message Passing. ICML 2023.

[3] Jin et al. Homomorphism Counts for Graph Neural Networks: All About That Basis. ICML 2024.

[4] Charilaos I. Kanatsoulis and Alejandro Ribeiro. Counting Graph Substructures with Graph Neural Networks. ICLR 2024.

**Theoretical Claims:**

I read and checked all proofs introduced. I am convinced Propositions 1 and 3 are correct. I think Proposition 2 is also correct, although I am less certain.

---

> ### Author Rebuttal · Authors · 2025-03-30
>
> Thank you for your thorough and thoughtful review. We are glad that you found the theoretical claims convincing and appreciated the clarity of our method and the rigor of our experimental setup.
>
> Bellow we report the answers to the comments and questions.
>
> **Q1. Aspect contributing to runtime of path mining algorithm**
>
> **R1.** In the cases considered here, where the number of nodes does not exceed a few hundred, the critical aspect is graph density.
> From the algorithmic point of view, the parameters that impact the most the computation time are the fraction of selected root nodes, $R$, the number of trials per depth $N$, and to a lesser extent the maximum DFS depth $D_\text{DFS}$, as can be seen in Figure 5. Although the worst-case complexity scales linearly with $D_\text{DFS}$ and $N$, their contributions are most of the time smaller: we suspect that the sparser the input graph, the lesser the contributions of $D_\text{DFS}$ and $N$ are (may it be at least as $N$ cannot exceed node degree), although this would require further and thorough analysis.
>
> **Q2: Similarity with Node2Vec**
>
> **R2**. A given DAG decomposition could be interpreted as an attempt to characterize the neighborhood of a root node in the sense of Node2Vec, in the limit case where the neighborhood is the whole graph. In this sense the two methods indeed share similarities. However, we characterize node pairs instead of single nodes, and rather than listing the neighboring nodes, we focus on the ordering in which the nodes were discovered by the search to unveil graph structures at various scales.
>
> **Q3. Choice of datasets used**
>
> **R3.** The practical reason that drove our choice of test datasets was that we looked for existing configurations for the training of GRIT and/or CSA, the two graph transformer methods implementing RWSE, in order to validate our methodology.
> The exploration of the benefits of SPSE on more diverse graph topologies will be an exciting future research direction.
>
> **Q4. Parameter study on $\alpha$, $\beta$ and $n$**
>
> **R4**
> Our analysis of the path encoding parameters (Equation 4) revealed two main factors contributing significantly to variations in model performance:
> * Compression of the dynamic range of path counts (controlled primarily by parameter $n$).
> * Adjustment of the final output range (affected by both parameters $\alpha$ and $n$).
>
> These effects are demonstrated in the tables below, presenting model accuracy for GPS on the PATTERN dataset and GRIT on the MNIST dataset, across various hyperparameter configurations:
>
> | PATTERN / GPS  |     |      $n$      |
> |----------------|:------:|:----------:|
> | $\alpha$       |    2   |      3     |
> |       0,2      | 86,815 | **86,834** |
> |       0,5      |    -   | 86,819     |
>
> | MNIST / GRIT |    |     $n$      |          |
> |--------------|:------:|:------:|:----------:|
> | $\alpha$     |    1   |    2   |      3     |
> |      0,5     | 98,189 | 98,224 | **98,294** |
>
> In these cases, the large path counts explain why the best results are obtained for a higher $n$ / lower $\alpha$.
>
> **Q5. Expressivity regarding WL / RWSE**
>
> **R5:**
> We provide below an initial discussion about the expressivity of the proposed SPSE encoding as an answer to this insightful comment.
> A reasoning similar to that employed for Path-WL [1] shows that composing $k$ global attention layers with SPSE results in a model that is strictly more expressive than $k$ iterations of the 1-WL color refinement algorithm.
> We note also that all properties regarding the expressivity of GRIT remain valid when SPSE replaces RWSE as the chosen structural encoding method.
> However, comparing the expressivity of SPSE and RWSE through isomorphism tests is more complex. Contrary to WL- / Path-trees introduced in [1] and [2], SPSE and RWSE aggregate information over simple paths and walks without enumerating them, which prevents one from using the strategy of the proof of Theorem 3.3 of [2] to draw any conclusion.
> A rigorous theoretical analysis is required to characterize the precise relationship between the two encodings within the WL hierarchy, which is left as future work.
>
> [1] Graziani et al., "The Expressive Power of Path-Based Graph Neural Networks." ICML, 2024.
>
> [2] Michel et al., “Path Neural Networks: Expressive and Accurate Graph Neural Networks”, ICML, 2023
>
> **Q6. Missing SEs benchmark**.
>
> **R6.** The missing reference will be added to the benchmark.

---

> > ### Comment · Reviewer_wLjL · 2025-04-04
> >
> > I am satisfied with these replies. I think the parameter studies and some discussion on the runtime of the path mining algorithm would be valuable to include in the final version/appendix. I stand by my initial assessment of this work and give my best wishes to the authors.

---

### Decision · Program_Chairs · 2025-05-01

**Decision:**

Accept (poster)

**Comment:**

This paper introduces Simple Path Structural Encoding (SPSE), a novel method for enhancing graph transformers by leveraging simple path counts instead of random walk probabilities, addressing limitations in capturing local graph structures. The reviewers unanimously recognized the theoretical and empirical contributions, with strong support for SPSE's improved expressivity, particularly in distinguishing cyclic patterns. While some concerns were raised about computational scalability and the need for further hyperparameter tuning, the authors provided thorough rebuttals, clarifying algorithmic details and proposing future directions for optimization. The experimental design was praised for its rigor, including significance testing and comparisons with RWSE across diverse benchmarks. Although performance gains were noted as modest in some cases, the method's novelty and potential for broader applications were deemed compelling. Given the paper's clear contributions, robust validation, and constructive reviewer feedback, we recommend acceptance to ICML 2025.